# Postbiotics for Preventing and Treating Common Infectious Diseases in Children: A Systematic Review

**DOI:** 10.3390/nu12020389

**Published:** 2020-01-31

**Authors:** Jeadran N. Malagón-Rojas, Anastasia Mantziari, Seppo Salminen, Hania Szajewska

**Affiliations:** 1Doctorado en Salud Pública, Facultad de Medicina, Universidad El Bosque, 110121 Bogota, Colombia; 2Instituto Nacional de Salud de Colombia, 111321 Bogota, Colombia; 3Functional Foods Forum, Faculty of Medicine, University of Turku, 20520 Turku, Finland; anastasia.mantziari@utu.fi (A.M.); sepsal@utu.fi (S.S.); 4Department of Paediatrics at the Medical University of Warsaw, 02091 Warsaw, Poland; hania@ipgate.pl

**Keywords:** 1 postbiotics, 2 probiotics, 3 common infectious diseases, 4 children, 5 public health

## Abstract

Postbiotics have recently been tentatively defined as bioactive compounds produced during a fermentation process (including microbial cells, cell constituents and metabolites) that supports health and/or wellbeing. Postbiotics are currently available in some infant formulas and fermented foods. We systematically reviewed evidence on postbiotics for preventing and treating common infectious diseases among children younger than 5 years. The PubMed, Embase, SpringerLink, and ScienceDirect databases were searched up to March 2019 for randomized controlled trials (RCTs) comparing postbiotics with placebo or no intervention. Seven RCTs involving 1740 children met the inclusion criteria. For therapeutic trials, supplementation with heat-killed *Lactobacillus acidophilus* LB reduced the duration of diarrhea (4 RCTs, *n =* 224, mean difference, MD, −20.31 h, 95% CI −27.06 to −13.57). For preventive trials, the pooled results from two RCTs (*n =* 537) showed that heat-inactivated *L. paracasei* CBA L74 versus placebo reduced the risk of diarrhea (relative risk, RR, 0.51, 95% CI 0.37–0.71), pharyngitis (RR 0.31, 95% CI 0.12–0.83) and laryngitis (RR 0.44, 95% CI 0.29–0.67). There is limited evidence to recommend the use of specific postbiotics for treating pediatric diarrhea and preventing common infectious diseases among children. Further studies are necessary to determine the effects of different postbiotics.

## 1. Introduction

Respiratory and gastrointestinal infections constitute a significant public health problem, especially for preschool children. Children younger than 5 years are especially vulnerable to infections; this predisposition is thought to be driven by a complex network of modulators, involving the immaturity of the immune response and organ function [1]. Since early 1980, the use of probiotics has been proposed to reduce the burden of common infectious disease among children [2,3,4,5] and infants [6]. However, a noteworthy part of the scientific community does not support probiotic interventions in younger children, due to rare case reports of probiotic-related infections such as bacteremia, necrotizing enterocolitis, pneumonia, and meningitis [7,8,9,10,11]. In addition, it has been stated that specific probiotic strains could express putative virulence factors, thus, increasing their tendency to adhere, invade, and produce cytotoxic effects [12]. Moreover, another concern is the possibility that they could transfer antibiotic resistance genes to pathogenic bacteria in the gut [13,14]. Because of these challenges, supplementation with postbiotics has been proposed as an alternative strategy to reduce the incidence of infectious diseases in children. Indeed, it has been suggested that viable and non-viable (heat-inactivated) *Lactobacillus rhamnosus* GG may have similar effects on the duration of diarrhea in patients with acute rotavirus diarrhea [15].

Currently, there is no globally accepted definition of postbiotics [16,17]. Nevertheless, in this review, in light of the lack of an acceptable definition, we will refer to postbiotics using their tentative definition as follows: Postbiotics are bioactive compounds produced during a fermentation process (including microbial cells, cell constituents and metabolites) that supports health and/or wellbeing [18].

The mechanisms of action of postbiotics are not well characterized. It has been postulated that inactivated probiotics or their components may modulate the immune response of the host. Stimulation of the immune system can be provoked by structural components of the bacterial pellicle, capsule, or cell wall constituents [19] such as peptidoglycans, liposaccharides [20], and S-layer proteins [21]. The initial host response is equipped with a series of pattern recognition receptors (PRRs). These PRRs are proteins largely expressed by cells of the innate immune system that recognize microbe-specific molecules called pathogen-associated molecular patterns (PAMPs). Two types of the PRRs have been proposed as playing principal roles in the regulation of the host’s innate immune response: the Toll-like receptors (TLRs) and the nucleotide-binding oligomerization domain (NOD)-like receptors (NLRs). The different types of TLRs can bind to specific microbial structures, including bacterial carbohydrates (lipopolysaccharides), bacterial nucleic acids (DNA or RNA), bacterial peptides (flagellin), lipoproteins, lipoteichoic acids, and peptidoglycans [22]. The NLRs are PRRs similar to TLRs associated with both the innate and acquired (adaptive) immune response [22]. The NLRs are able to recognize different ligands from microbial pathogens such as viral RNA, peptidoglycans, and flagellin among others [23]. Additionally, it has been proposed that the NLRs may respond to different cytokines, including the interferons [24]. In that regard, NLRs may contribute to the activation of T and B cells in response to the postbiotic stimuli. Components from postbiotic products, such as lipoteichoic acid, could have an anti-inflammatory effect, reducing the production of reactive oxygen species (ROS) and IL-2, and increasing the production of cytokines IL-4, IL-6, and IL-10 [25].

All in all, postbiotics seem to mimic the beneficial therapeutic effects of probiotics, while avoiding the risk of administering live microorganisms, especially in high-risk populations such as children younger than the age of 5 years [26,27]. The present systematic review aims to update the evidence on the effectiveness of postbiotics for the prevention and treatment of common infectious diseases in young children.

## 2. Material and Methods

The guidelines from the Cochrane Collaboration for undertaking and reporting the results of a systematic review [28] and the Preferred Reporting Items for Systematic Reviews and Meta-Analyses (PRISMA) guidelines were followed [29].

### 2.1. Types of Studies

Only randomized clinical trials were included.

### 2.2. Type of Participants

Participants had to be young children (preferably younger than 5 years of age).

### 2.3. Types of Interventions

In the intervention group, participants received postbiotics (as defined in the Introduction). The participants in the control group received placebo or no intervention.

### 2.4. Types of Outcome Measures

All clinical outcomes reported by the investigators, if relevant to the current review, were considered. However, for therapeutic trials, our primary focus was on the duration and/or the severity of the disease episode. For preventive trials, our primary focus was on the disease occurrence and number of episodes of diseases. For all trials, we also evaluated adverse effects. Non-clinical outcomes were not considered.

### 2.5. Search Methods

The following databases were included in the search: PubMed, Embase, SpringerLink, and ScienceDirect, from 1970 to March 2019, without language restrictions. Moreover, other registered clinical trials from the ClinicalTrials.gov website [30], EU Clinical Trials Register website [31] and PROSPERO [32] were searched. Two of the researchers designed the search algorithms based on the following MeSH terms: #postbiotic, #metabiotic, #paraprobiotic, #infants, #diarrhea, #respiratory tract infection. The selected search algorithm was the following: (((((postbiotic[All Fields] OR postbiotics[All Fields] OR postbiotics’[All Fields]) OR (metabiotic[All Fields] OR metabiotics[All Fields])) OR (paraprobiotic[All Fields] OR paraprobiotic’[All Fields] OR paraprobiotics[All Fields])) OR ((“cells”[MeSH Terms] OR “cells”[All Fields] OR “cell”[All Fields]) AND free[All Fields] AND supernatant[All Fields])) OR CFS[All Fields]) AND (“infant”[MeSH Terms] OR “infant”[All Fields] OR “infants”[All Fields]) AND “humans”[MeSH Terms]. The search algorithm was adjusted to the structure of each database.

The following sources of information were excluded: literature reviews, book chapters, conference proceedings, blogs, newspaper articles and letters to the editor, articles that referred to technologies (including live probiotics vs. placebo), and duplicate documents.

Two of the researchers downloaded the comma-separated values (CSV) file generated by each database. Later, one of the reviewers uploaded the files to Rayyan^®^ web application for systematic reviews [33].

### 2.6. Selection of Articles

Rayyan^®^ allowed the reviewers to blind the selection of articles. Titles and abstracts were evaluated by two observers, who applied the inclusion and exclusion criteria checklist. In case of disagreement, a third blind evaluator settled the differences between the researchers. The PRISMA diagram [29] was used to guide the review process.

### 2.7. Data Extraction and Management

Information related to principal author, publication year, type of population, age, intervention and control groups, outcomes measured, and definitions of the primary outcome were extracted.

### 2.8. Assessment of Risk of Bias in Included Studies

Two of the researchers assessed the risk of bias of the studies that met the inclusion criteria. The Cochrane Collaboration’s tool for assessing risk of bias was used. The risk of bias parameters includes the type of randomization method (selection bias), allocation concealment (selection bias), blinding of participants and personnel (performance bias), blinding of outcome assessment (detection bias), incomplete outcome data (attrition bias), and selective reporting (reporting bias). In all cases, a positive answer (+) was considered to low risk of bias; a negative answer (-) indicated a high risk of bias.

### 2.9. Measures of Treatment Effect

The effect measures included the risk ratio (RR) for dichotomous data and the mean difference (MD) for continuous data. Both were evaluated with 95% confidence intervals (95% CI). If appropriate, the number needed to treat (NNT) was calculated as the inverse of the pooled absolute risk differences between two treatment options and expressed with 95% CI.

### 2.10. Dealing with Missing Data

In the study by Boulloche et al. (1994), missing standard deviations were obtained from a p value by the method described in the Cochrane Handbook for Systematic Reviews of Interventions [34].

### 2.11. Assessment of Heterogeneity

The percentage of the total variation between studies, i.e., the heterogeneity, was quantified by Chi^2^ and *I*^2^. The resultant *I*^2^ represents the percentage of the total variation between studies that is attributable to heterogeneity rather than to chance. A heterogeneity >50% was considered as high variability among the studies [34].

### 2.12. Data Synthesis

The reviewers performed data extraction. In the case of dichotomous outcomes, the number of participants who experienced the event and total number of participants were extracted. For continuous outcomes, the total number of participants, means, and standard deviations were extracted.

Nocerino et al. [35] included two intervention arms: one with fermented milk and the other with a fermented rice product (active products were cow’s milk fermented by *L. acidophilus* LB or rice fermented by *L. acidophilus* LB). In this case, the data were pooled into one arm. If other comparisons were made, these other arms were not evaluated here. For example, Boulloche et al. [36] compared *L. acidophilus* LB with loperamide and placebo. The data about the loperamide arm were not included.

For the analysis of dichotomous and continuous outcomes, the software Review Manager ^®^ (RevMan, computer program, version 5.3. Copenhagen: The Nordic Cochrane Centre, The Cochrane Collaboration, 2014) was used. All pooled analyses were based on the random effect model.

## 3. Results

See Figure 1 for a search flow diagram. A total of 144 documents was obtained from the online search, and three additional articles were suggested by one of the authors. After the removal of 80 duplicates, 67 articles remained. These 67 remaining articles were screened, resulting in the exclusion of 46 more records. Twenty-one full-text articles were assessed for eligibility. Fourteen of these articles were excluded for not meeting the inclusion criteria, resulting in seven articles included in the qualitative and quantitative analysis.

Characteristics of the seven included trials involving 1740 participants are summarized in Table 1. Principal characteristics of the analysed studies. Four of the trials were carried out in Europe; two in Latin America; and one in Thailand. The ages of the children enrolled in the trials ranged from 1 to 48 months. The interventions studied included heat-killed *L. acidophilus* LB (4 RCTs) [36,37,38,39], fermented infant formula containing “non-live” *Bifidobacterium breve* C50 *& Streptococcus thermophilus* 065 (1 RCT) [40], and lyophilized heat-killed *L. paracasei* CBA L74 administered in cow’s milk or rice (2 RCTs)[35,41]. Heat was the mode of inactivation for the killed bacteria administered in all the included studies. Also, three studies reported that the sachets administered to the intervention groups contained the heat-killed bacteria and a neutralized supernatant of spent culture medium [37,38,39].

There were no identified registries of current ongoing, prematurely ended, suspended clinical trials in the European and United States databases. Also, there were not any registered systematic reviews on PROSPERO about the use of non-live bacteria in the prevention of common infectious diseases in children younger than 5 years.

### 3.1. Bias Assessment

Risk of bias in the included studies is presented in Figure 2. All of the included trials were described at least as double-blinded (participants and personnel). Withdrawals and dropouts were adequately described in the majority of the trials. However, in most of therapeutic trials, the risk of reporting bias was unclear.

### 3.2. Summary of Findings

#### 3.2.1. Therapeutic Trials (Gastroenteritis)

A total of 5 RCTs were included.

##### Heat-Killed L. Acidophilus LB

There were a total of four RCTs using non-viable *L. acidophilus* LB for the treatment of acute diarrhea in 224 children from different locations [36,37,38,39]. The average time of treatment was 4.3 days (SD = 0.47). Compared with the placebo group, the *L. acidophilus* LB group had a significant reduction in the duration of diarrhea episodes (4 RCTs, *n =* 224, MD −20.31 h, 95% CI −27.06 to −13.57). The heterogeneity was not significant (I^2^ = 24%) (Figure 3).

##### Fermented Formula with B. Breve C50 and Str. Thermophilus 065

One randomized, double-blind, placebo-controlled trial was carried out in 94 centers in France [36]. This study reported that there was no significant difference between the postbiotic and placebo groups in the duration of the diarrhea episodes (*n =* 913, MD 2.64 h, 95% CI −6.42 to 11.70) (Figure 3). Nevertheless, the authors reported fewer cases of dehydration in the group that received fermented formula with *B. breve* C50 and *Str. thermophilus* 065 (RR 0.41 95% CI 0.20 to 0.82).

#### 3.2.2. Prevention Trials

Three prevention trials involving children aged 4 and 48 months [35,40,41] were included. All trials were carried out in Europe. Three of the studies reported data on the prevention of gastroenteritis (diarrhea) as an outcome, while two of them also included data on the prevention of respiratory tract infections [31,37]. The studies involved the use of inactivated, heat-killed *L. paracasei* CBA L74 [31,37] and “non-live” *B. breve* C50 and *Str. Thermophiles* [37].

(1) Gastroenteritis

##### Heat-Killed *L. Paracasei* CBA L74

Two RCTs [35,41] assessed the effect of the administration of heat-killed *L. paracasei* CBA L74 for 90 days in preventing diarrhea in 537 otherwise healthy Italian children aged 12 to 48 months. Compared with the placebo group, in the heat-killed *L. paracasei* group there was a significant reduction in the number of episodes of diarrhea (2 RCTs, *n* = 537, RR 0.51, 95% CI 0.37 to 0.71). No heterogeneity was found (I^2^ = 0%) (Figure 4).

##### Fermented Formula with *B. Breve* C50 (BbC50) and *Str. Thermophilus* 065

One RCT assessed the effect of a daily administration of a fermented formula with non-live *B. breve* C50 and *Str. thermophilus* 065 for 5 months in preventing diarrhea in 913 otherwise healthy French infants aged 4 to 6 months [40]. There was no difference in the number of episodes of diarrhea between the placebo group and the intervention group (*n =* 913, RR 1.01, 95% CI 0.98 to 1.04) (Figure 4).

(2) Respiratory Tract Infections

##### Heat-Inactivated *L. Paracasei* CBA L74

Two RCTs [35,41] assessed the effect of the administration of heat-killed *L. paracasei* CBA L74 for 90 days in preventing respiratory tract infections in 537 healthy Italian children. The pooled analysis of these two trials demonstrated that the use of postbiotics reduced the number of cases of pharyngitis (RR 0.31, 95% CI 0.12 to 0.83; NNT 2.4, 95% CI 1.9 to 3.1), laryngitis (RR 0.44, 95% CI 0.29 to 0.67; NNT 3.6, 95% CI 2.5 to 6.5), and tracheitis (RR 0.65, 95% CI 0.50 to 0.86; NNT 8.05, 95% CI 4.5 to 32.4). No heterogeneity was found. There was no significant difference in the risk of rhinitis (RR 0.77, 95% CI 0.57 to 1.03; I^2^ = 0%) and otitis media between the postbiotic and placebo groups (RR 0.36, 95% CI 0.13 to 1.02; I^2^ = 69%) (Figure 5).

### 3.3. Adverse Effects

Only three of the included RCTs evaluated the secondary effects of postbiotics [35,38,41]. In these trials, there were no significant differences between the experimental and control groups in regard to adverse effects.

## 4. Discussion

This systematic review aimed to update current evidence on the benefits of using postbiotics for preventing and treating common infectious diseases among children under the age of 5 years. Only seven trials met the inclusion criteria. In the therapeutic trials, the administration of a heat-killed *L. acidophilus* LB reduced the duration of diarrhea compared with a placebo. Supplementation with fermented formula with non-viable *B. breve* C50 and *Str. thermophilus* 065 had no effect on the duration of gastrointestinal infections/diarrheal episodes in children younger than 5 years. In the preventive trials, the administration of heat-killed *L. paracasei* CBA L74 reduced both the number of cases of acute gastroenteritis and respiratory tract infections among children younger than 5 years.

There were no significant differences in adverse effects between the intervention and placebo groups. Only one RCT reported severe dehydration associated with the heat-killed *L. acidophilus* LB administration. Such side effects have been reported previously in an RCT that compared the administration of micronutrients vs. lyophilized, heat-killed, *L. acidophilus* LB + micronutrients vs. placebo [42]. The authors reported a higher rate of vomiting and abdominal distention in the group that received heat-killed *L. acidophilus* LB + micronutrients. It seems that heat-inactivated lactic acid bacteria may activate immune inflammatory mechanisms in children, related to enterocyte adhesion and proinflammatory chemokines [43]. The remaining trials did not report on adverse events.

There are a few works that have reported potential side effects of postbiotics administration. One rapid review carried out to evaluate the effect of acidified and fermented infant formulas on the development of clinical symptoms of D-lactic acidosis in healthy infants and children. The authors suggested a theoretical possibility of developing subclinical accumulation of D-lactate. Nevertheless, the review found that a few cases in non-healthy infants (e.g., short bowel syndrome) fed with acidified formulas developed pediatric D-lactic acidosis [44]. Another narrative review that focused on identifying potential applications of postbiotic supplementation in early life reported no significant side effects in healthy children [45].

Despite the availability of other sorts of inactivation modes (e.g., chemicals, sonification, ultra violet (U.V.) radiation) [17,19], heat treatment seems to be the preferred method for inactivating probiotics [46]. However, some authors have suggested that inactivation by U.V. radiation is the most appropriate method for studying non-viable probiotics and preparing control products [47]. Nevertheless, there are no data to allow one to reach conclusions on the final effects of various methods of inactivation.

### 4.1. Limitations

As the definition of postbiotics is still disputed, only the working definition of postbiotics was used. We only included randomized clinical trials, which implies a limited number of studies. Even though only randomized trials were included, the methodological quality of the included studies was often questionable. This was especially true for studies carried out before 2010, i.e., the year when the CONSORT statement was introduced, contributing to better reporting of RCT results. Potential limitations of the included trials were unclear random sequence generation, unclear allocation concealment, and unclear blinding of outcome assessment in some of the trials. Most of the evidence from both preventive and therapeutic RCTs comes from Europe. The lack of information from different locations may reduce the generalizability of the results. This is important when considering that none of the reported effects have been confirmed in repeated studies.

Although one goal was to measure the impact of the therapeutic interventions with different postbiotics on dehydration or diarrhea severity, it was not possible to compare these outcomes. First, because of the dissimilarities in the inclusion criteria among the studies, some of them included children with mild and moderate dehydration, while others rejected the inclusion of children with any grade of dehydration. Also, none of the RCTs reported on the recovery from dehydration during the treatment.

### 4.2. Comparison with Previous Reports

The effects of heat-inactivated *L. acidophilus* LB were previously evaluated by Szajewska et al. [48]. In line with the current report, they found that the use of heat-killed *L. acidophilus* LB compared with placebo reduced the duration of diarrhea associated with acute gastroenteritis in hospitalized children. Most of the current evidence about the use of heat-killed *L. acidophilus* LB supplementation for the acute management of diarrhea comes from studies carried out between 1994 and 2007. No new studies were identified. Of note, the use of *L acidophilus* LB for the management of acute gastroenteritis is in line with current recommendations [49,50].

The effects of fermented infant formula containing *B. breve* C50 & *Str. thermophilus* was earlier evaluated by Szajewska et al. [51,52]. The authors concluded that the use of fermented infant formula, compared with the use of standard infant formula, does not offer clear additional benefits, although some benefit on gastrointestinal symptoms cannot be excluded. In the search for the current review, no new studies were identified.

## 5. Conclusions

Postbiotics, obviating the need to preserve viability of the bacteria, while maintaining the benefits in terms of reducing the number of cases of common infectious diseases in young children, could be an interesting alternative for regions where probiotic administration is not easy. There is only limited evidence to recommend the use of specific postbiotics for treating pediatric diarrhea and preventing cases of common infectious diseases among children; specifically, heat-killed *L. acidophilus* LB for the management of acute diarrhea and heat-killed *L. paracasei* CBA L74 for preventing gastrointestinal and respiratory tract infections. Further studies should be carried out in different locations, as well as include a cost efficacy analysis. Formulating an official scientific definition for postbiotics will enable the design of better future RCT studies.

## Figures and Tables

**Figure 1 nutrients-12-00389-f001:**
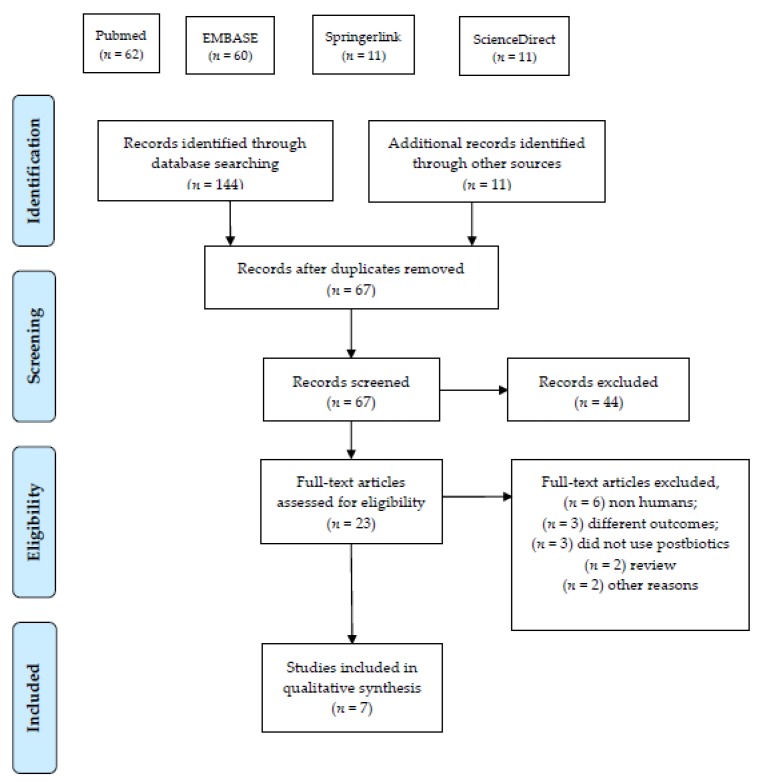
Identification process for eligible trials.

**Figure 2 nutrients-12-00389-f002:**
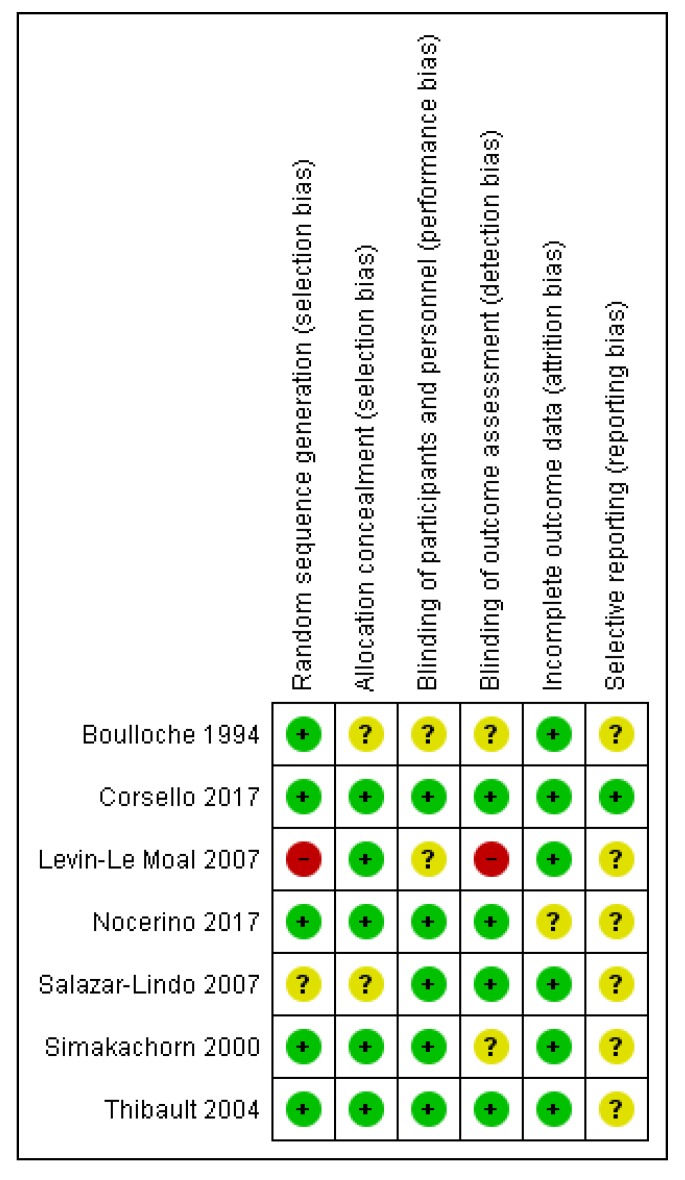
Risk of bias in the indclued studies.

**Figure 3 nutrients-12-00389-f003:**
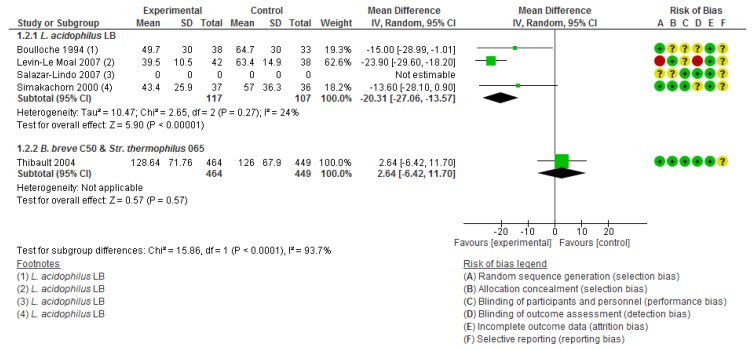
Effect of postbiotic supplementation on the duration (hours) of acute diarrhoea.

**Figure 4 nutrients-12-00389-f004:**
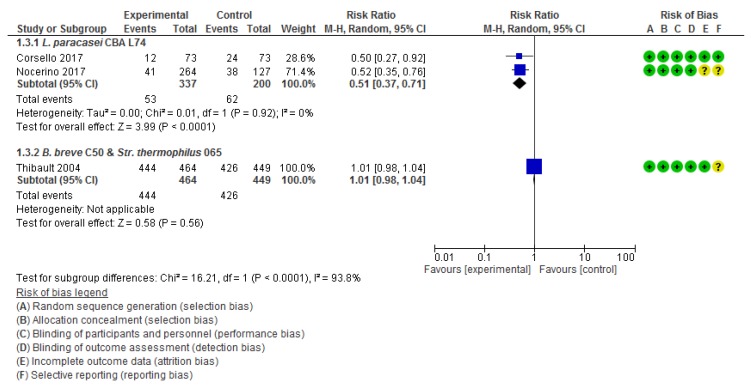
Effect of postbiotic supplementation on preventing gastroenteritis.

**Figure 5 nutrients-12-00389-f005:**
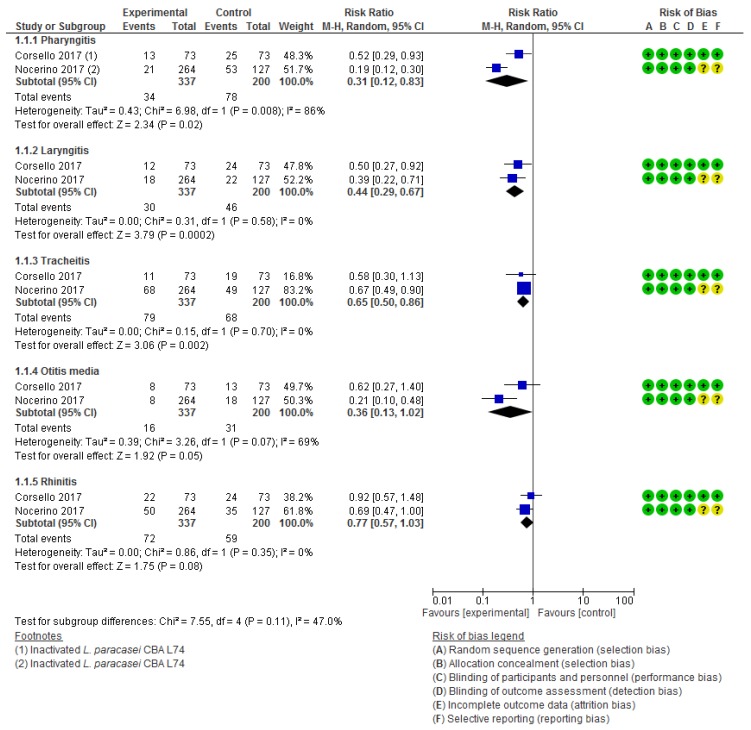
Effect of postbiotic supplementation (heat-inactivated L. paracasei CBA L74) on preventing respiratory tract infections.

**Table 1 nutrients-12-00389-t001:** Principal characteristics of the analysed studies.

Author (Year, Country)	Age	Population (*n*)	Intervention and Control Group	Duration of the Intervention	Primary Outcome Measured	Definition of the Primary Outcome
Boulloche J, 1994 (France) [36]	1 to 48 mo	Acute diarrhoea (*n =* 71)	Heat killed *L. acidophilus* LB vs placebo	4 days	Duration of diarrhoea	The first normal stool.
Simakachorn (Thailand) N,2000 [39]	3 to 24 mo	Acute diarrhoea without severe dehydration (*n =* 73)	Lyophilized heat-killed *L. acidophilus* LB vs placebo	5 days	Diarrhoea duration	The end of the diarrhoea was defined as two consecutive well-formed stools followed an unformed stool or when no stool was passed for 12 h.
Thibault H, 2004 (France) [40]	4 to 6 mo	Healthy infants (*n =* 913)	Fermented formula with heat-killed *Bifidobacterium breve C50 and Streptococcus thermophilus* vs placebo	5 months	Number of acute diarrhoea episodes Diarrhoea duration	Duration was defined as the time passed between the first diarrhoea episode and when the stools were formed.
Salazar-Lindo, 2007 (Peru) [38]	3 to 48 mo	Acute diarrhoea (less than 3 days) (*n =* 80)	Heat killed *Lactobacillus* LB vs placebo	4,5 days	Duration of diarrhoea caused by non-rotavirus	The time to the first normal stool followed by 2 consecutive normal stools
Levin-Le Moal, V., 2007 (Ecuador) [37]	1 to 12 mo	Acute diarrhoea (*n =* 80)	Heat killed *L. acidophilus* LB vs placebo	4 days	Duration of the diarrhoea episode	Time passed between the first diarrhoea episode and when the stools were formed.
Nocerino R, 2017 (Italy) [35]	12 to 48 mo	Healthy children attending to day care or preschool at least five days a week (*n =* 377)	Cow’s milk or rice with fermented milk with *L. paracasei* CBA L74 and inactivated vs placebo	3 months	Proportion of children experiencing at least one episode of common infectious disease	Diarrhoea (the presence of 3 episodes of soft/liquid faeces in 24 h with or without fever or vomiting). Upper respiratory tract infections (the occurrence of 1 respiratory symptom(s) (runny nose, cough, sore throat, aphony, shortness of breath, otalgia, otorrhoea, extroversion of tympanic membrane with or without hyperaemia) in the absence or presence of 1 systemic symptom(s).
Corsello G, 2017 (Italy) [41]	12 to 48 mo	Healthy children, attending day care or preschool for at least 5 days a week (*n =* 146).	Lyophilized heat-killed *L. paracasei* CBA L74 vs placebo	3 months	The rate of children experiencing at least one episode of Common infectious disease	Laryngitis: inspiratory wheezing with cough and hoarse voice with or without chest indrawing, stridor, aphony and fever. Pharyngitis: inflammation of the pharyngeal tonsils that may be accompanied by other nonspecific symptoms. Tracheitis: symptoms of airway obstruction or impending respiratory failure or both (including cough, mucus production, shortness of breath, or fever). Acute otitis media presence of tympanic membrane inflammation and by the presence of otalgia or, otorrhoea, irritability, and fever).

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
