# Peer review of "Postbiotics for Preventing and Treating Common Infectious Diseases in Children: A Systematic Review"

_nutrients, 2020, doi:10.3390/nu12020389_

Round 1

Reviewer 1 Report

The present work is dealing with the health effects of postbiotic in clinical trials, focusing on the common infectious diseases in children. The systematic review is well designed and performed. The weakness is the low number of eligible studies, which raises doubts about the significance of this review. The merit is to highlight the scarcity of clinical trials well designed.

Minor revision:

P4 L63: Why a universally acceptable definition is lacking?

P4 L64-66: Why did you choose this definition? Please, provide a reference.

Author Response

Response to reviewer 1 comments

Comment 1. The present work is dealing with the health effects of postbiotic in clinical trials, focusing on the common infectious diseases in children. The systematic review is well designed and performed.

Response: We thank the Reviewer for these kind words.

Comment 2. The weakness is the low number of eligible studies, which raises doubts about the significance of this review. The merit is to highlight the scarcity of clinical trials well designed.

Response: We agree with the Reviewer. However, we believe that our demonstration of clinical uncertainty about this issue is an important finding. As pointed out by Alderson and Roberts (BMJ 2000}, clinical uncertainty is a prerequisite for the large-scale RCTs needed to evaluate the influence of such interventions; it also helps to clarify available treatment options and stimulate new and better research.

Ref. Alderson P, Roberts I. Should journals publish systematic reviews that find no evidence to guide practice? Examples from injury research. BMJ. 2000;320(7231):376–377. doi:10.1136/bmj.320.7231.376

Minor revision:

Reviewer question 1;  P4 L63: Why a universally acceptable definition is lacking?

Response: So far research has been mainly focused on probiotics, live microbial food supplements, and only recently scientific evidence has emerged indicating that inactivated probiotics and their metabolites can also have health effects. Additionally, the European Union Safety Assessment system for microbes causes challenges for live bacteria and it is easier to have an inactivated microbe approved for food use. Therefore, several organizations are working on the term and the International Scientific Association of Probiotics and Prebiotics has organized a consensus conference on the term.

Reviewer question 2: P4 L64-66: Why did you choose this definition? Please, provide a reference.

Response: recently, Collado et al discussed that an expert consensus on the exact definition of postbiotics is needed mostly due to regulatory issues. They provided a new definition: https://doi.org/10.3920/BM2019.0015.

We chose the one reference that was most recent and most comprehensive at the time of writing the manuscript. The reference is:

Redefinition: Collado Mc, Vinderola G, Salminen S. Postbiotics, facts and open questions. A position paper on the need for a consensus definition. Beneficial Microbes 2019 10: 711-719.

Reviewer 2 Report

The manuscript reports about an extensive meta-analysis perfomed on the available RCT to investigate the efficacy and safety of postbiotics supplemenation to prevent or treat gastroenteric or respiratory infections in  children whose age is comprised from 0 to 48 months. Overall the paper is well designed and performed: the major limitation is the very smal number of RCT that were evalueted due to the scarcity of these studies in the literature. The analysis performed is anything that conclusive in term of efficiency and safety (this last feature is likely less evaluated in the RCTs included)  of postbiotics treatment in childhood. The statistical work is well performed even in the absence of large amount of data.  The Authors should rethink to their discussion section since this is presently more or less a repeated summary of the findings described in the appropriate section of the manuscript: my suggestion is to make it shorter and more focused obn the many limitation of the study given the small number of RCTs evaluated. This is partially done in the paragraph "limitations" but I believe this must be extensively underlined since no final conclusion can be overtaken at the end of the study itself.

Author Response

Response to reviewer 2 comments

Comment 1. “The manuscript reports about an extensive meta-analysis performed on the available RCT to investigate the efficacy and safety of postbiotics supplementation to prevent or treat gastroenteric or respiratory infections in children whose age is comprised from 0 to 48 months. Overall the paper is well designed and performed”:

RESPONSE: We thank the Reviewer for these kind words.

Comment 2. The major limitation is the very small number of RCT that were evaluated due to the scarcity of these studies in the literature. The analysis performed is anything that conclusive in terms of efficiency and safety (this last feature is likely less evaluated in the RCTs included) of postbiotics treatment in childhood.

RESPONSE: We agree with these comments. However, as we stated in our response to Reviewer #1, the demonstration of clinical uncertainty is a prerequisite for the large-scale RCTs needed to evaluate the influence of such interventions; it also helps to clarify available treatment options and stimulate new and better research.

Comment 3. The statistical work is well performed even in the absence of a large amount of data.  The Authors should rethink their discussion section since this is presently more or less a repeated summary of the findings described in the appropriate section of the manuscript: my suggestion is to make it shorter and more focused on the many limitations of the study given the small number of RCTs evaluated. This is partially done in the paragraph "limitations" but I believe this must be extensively underlined since no final conclusion can be overtaken at the end of the study itself.

RESPONSE:  As suggested by the Reviewer, we did our best. Nevertheless, we have now shortened the discussion slightly and modified it according to reviewer comments and the corrected on is presented below:

Discussion

This systematic review aimed to update current evidence on the benefits of using postbiotics for preventing and treating common infectious diseases among children under the age of 5 years. Only 7 trials met the inclusion criteria. In the therapeutic trials, the administration of a heat-killed L. acidophilus LB reduced the duration of diarrhea compared with a placebo. Supplementation with fermented formula with non-viable B. breve C50 and Str. thermophilus 065 had no effect on the duration of gastrointestinal infections/diarrheal episodes in children younger than 5 years. In the preventive trials, the administration of heat-killed L. paracasei CBA L74 reduced both the number of cases of acute gastroenteritis and respiratory tract infections among children younger than 5 years.

There were no significant differences in adverse effects between the intervention and placebo groups. Only one RCT reported severe dehydration associated with the heat-killed L. acidophilus LB administration. Such side effects have been reported previously in an RCT that compared the administration of micronutrients vs. lyophilized, heat-killed, L. acidophilus LB + micronutrients vs. placebo (38). The authors reported a higher rate of vomiting and abdominal distention in the group that received heat-killed L. acidophilus LB + micronutrients. It seems that heat-inactivated lactic acid bacteria may activate immune-inflammatory mechanisms in children, related to enterocyte adhesion and proinflammatory chemokines (39). The remaining trials did not report on adverse events.

There are a few works that have reported potential side effects of postbiotics administration. One rapid review carried out to evaluate the effect of acidified and fermented infant formulas on the development of clinical symptoms of D-lactic acidosis in healthy infants and children. The authors suggested a theoretical possibility of developing subclinical accumulation of D-lactate. Nevertheless, the review found that a few cases in non-healthy infants (e.g. short bowel syndrome) fed with acidified formulas developed pediatric D-lactic acidosis (40). Another narrative review focused on identifying potential applications of postbiotic supplementation in early life reported no significant side effects in healthy children(41).

Despite the availability of other sorts of inactivation modes (e.g., chemicals, sonification, ultraviolet (U.V.) radiation) (17), heat treatment seems to be the preferred method for inactivating probiotics (42). However, some authors have suggested that inactivation by U.V. radiation is the most appropriate method for studying non-viable probiotics and preparing control products (43). Nevertheless, there are no data to allow one to reach conclusions on the final effects of various methods of inactivation.

Limitations

As the definition of postbiotics is still disputed, only the working definition of postbiotics was used. We only included randomized clinical trials, which implies a limited number of studies. Even though only randomized trials were included, the methodological quality of the included studies was often questionable. This was especially true for studies carried out before 2010, i.e., the year when the CONSORT statement was introduced, contributing to better reporting of RCT results. Potential limitations of included trials were unclear random sequence generation, unclear allocation concealment, and unclear blinding of outcome assessment in some of the trials. Most of the evidence from both preventive and therapeutic RCTs come from Europe. The lack of information from different locations may reduce the generalizability of the results. This is important when considering that none of the reported effects have been confirmed in repeated studies.

Although one goal was to measure the impact of the therapeutic interventions with different postbiotics on dehydration or diarrhea severity, it was not possible to compare these outcomes. First, because of the dissimilarities in the inclusion criteria among the studies. Some of them included children with mild and moderate dehydration, while others rejected the inclusion of children with any grade of dehydration. Also, none of the RCTs reported on the recovery from dehydration during the treatment.
